# Systemic Chronic Treatment with Cannabidiol in Carioca High- and Low-Conditioned Freezing Rats in the Neuropathic Pain Model: Evaluation of Pain Sensitivity

**DOI:** 10.3390/ph16071003

**Published:** 2023-07-14

**Authors:** Carolina Macêdo-Souza, Silvia Soares Maisonnette, Jaime E. Hallak, José A. Crippa, Antônio W. Zuardi, J. Landeira-Fernandez, Christie Ramos Andrade Leite-Panissi

**Affiliations:** 1Department of Psychology, Faculty of Philosophy, Science and Letters of Ribeirão Preto, University of São Paulo, Ribeirão Preto 14040-901, Brazil; carolinamacedos@usp.br; 2National Institute for Translational Medicine (INCT-TM; CNPq), São Paulo 14049-900, Brazil; jhallak@fmrp.usp.br (J.E.H.); jcrippa@fmrp.usp.br (J.A.C.); awzuardi@fmrp.usp.br (A.W.Z.); 3Department of Psychology, Pontifical Catholic University of Rio de Janeiro, Rio de Janeiro 22451-900, Brazil; ssmaison@yahoo.com.br (S.S.M.); landeira@puc-rio.br (J.L.-F.); 4Department of Neuroscience and Behavioral Sciences, Ribeirão Preto School of Medicine, University of São Paulo, Ribeirão Preto 14049-900, Brazil

**Keywords:** animal model, anxiety, cannabidiol, chronic pain, GAD, neuropathic pain

## Abstract

Studies have shown high comorbidity of anxiety disorder and chronic pain; generalized anxiety disorder (GAD) and neuropathic pain are among these pathologies. Cannabidiol (CBD) has been considered a promising treatment for these conditions. This study investigated whether chronic systemic treatment with CBD alters pain in high- (CHF) and low-freezing (CLF) Carioca rats (GAD model) and control rats (CTL) submitted to chronic neuropathic pain. The rats were evaluated in the sensory aspects (von Frey, acetone, and hot plate tests) before the chronic constriction injury of the ischiatic nerve (CCI) or not (SHAM) and on days 13 and 23 after surgery. Chronic treatment with CBD (5 mg/kg daily) was used for ten days, starting the 14th day after surgery. The open field test on the 22nd also evaluated locomotion and anxiety-like behavior. CBD treatment had an anti-allodynic effect on the mechanical and thermal threshold in all lineages; however, these effects were lower in the CHF and CLF lineages. Considering emotional evaluation, we observed an anxiolytic effect in CTL+CCI and CHF+CCI after CBD treatment and increased mobility in CLF+SHAM rats. These results suggest that the CBD mechanical anti-allodynic and emotional effects can depend on anxiety level.

## 1. Introduction

Epidemiological studies have indicated that generalized anxiety disorder (GAD) and chronic pain are highly prevalent and debilitating diseases worldwide [1,2,3,4]. The annual investment costs in research and treatment of patients with GAD are high, associated with various personal, economic, and social commitments [3]. Similarly, it can also occur in chronic pain conditions. GAD is mainly characterized by a persistent and excessive state of worry [5], while chronic pain is marked by classes of pain that persist or recur after the recovery phase of an injury [6,7]. Concurrent with this prevalence, chronic or persistent pain is related to several psychiatric disorders [8]. In addition, several studies showed that among anxiety disorders, patients with GAD are the most susceptible to developing chronic pain [9,10].

Following the classification of chronic pain, neuropathic pain is caused by an injury or disease of the nervous system, which can be spontaneous or evoked, resulting in hyperalgesia and allodynia [8]. Concerning nociception modulation systems, the cannabinoid system is a constituent of the pain modulation process in its sensory-discriminative and cognitive-motivational aspects [11,12]. Studies suggest that cannabidiol (CBD) has a relevant role in pain modulation [11,13] and the modulation of the emotional system involved in different psychiatric disorders [14,15,16,17], likewise in anxiety disorders [18].

CBD is a promising strategy for treating pathological anxiety and chronic pain in a comorbid scenario. There is a growing body of evidence about the neurobiological mechanisms responsible for triggering the processes of this relationship between pain and anxiety [19,20]. However, the studies have shown that the independent systems corresponding to these processes have different psychophysiological aspects [7,21,22], and this interaction remains unclear.

Interestingly, in humans or non-humans, the perception and sensitivity of pain can be different and the level of anxiety can distinctly influence the outcome of pain management [23,24,25]. The pain modulation process is associated with mechanisms of descendent inhibition, and neurotransmitters such as serotonin, dopamine, and norepinephrine are present in this system, which control endogenous analgesia [26,27,28]. Additionally, anxiety levels may influence the expression of this pain control process through painful experiences and learning [29,30]. Therefore, other studies have suggested that high anxiety levels increase nociceptive sensitization and pain perception [19,31].

Animal models are used on a large scale for their validity and reliability [32,33]. Among the animal models of anxiety [34], Carioca high- and low-conditioned freezing rats (CHF and CLF, respectively) are the lineages selected from the phenotype of freezing response to contextual fear conditioning [35]. These lineages were developed through selection at the Department of Psychology at the Pontifical Catholic University of Rio de Janeiro in 2006. The protocol for selection was based on mating male and female albino Wistar rats with the highest and lowest conditioned freezing in response to the contextual cues of the experimental chamber, where animals were exposed to three unsignaled electric footshocks on the previous day. In this protocol, the difference in conditioned freezing was consistent in the third generation, with male rats exhibiting a more consistent freezing response than females. This animal model shows a stable anxiety trait across generations [36]. This GAD model [36] has been used to investigate comorbid conditions in this psychiatric condition [37,38,39,40]. However, pain sensitivity has not yet been studied in this model. We hypothesized that chronic pain modulation could be different depending on anxiety level. We tested this hypothesis by investigating whether chronic systemic treatment with CBD modulates pain sensitivity in rats from an animal anxiety model submitted to a neuropathic pain model.

## 2. Results

The sensitivity of CCI and SHAM in Carioca’s lineages was evaluated through nociceptive tests (von Frey, acetone evaporation test, and hot plate) on the baseline and on 13 and 23 days after surgery. Chronic CBD treatment took 10 days, from the 14th day after surgery until the 23rd, and the OFT was held on the 22nd after surgery. Sphericity was assumed in all analyses. The experimental timeline is depicted in Figure 1.

### 2.1. Mechanical Sensitivity Assessment

The present results showed that CBD reduced CCI-induced mechanical allodynia regardless of lineages; however, this effect was lower in CHF and CLF lineages (Figure 2A,C). Repeated measures analysis of variance (ANOVA) applied to all experimental groups revealed a significant difference in lineages, conditions, treatment, time, and interaction (Table 1). The Bonferroni post hoc test showed a significant difference (*p* < 0.0001) in the mechanical threshold on the 13th day in the CCI groups (CCI+VEHI and CCI+CBD) compared to the respective SHAM groups (Figure 2A–C) and baselines values. Chronic treatment with CBD reduced this mechanical allodynia in all strains, as the mechanical threshold on the 23rd day in the CCI+VEHI groups was different (*p* < 0.0001, Bonferroni test) when compared to the other groups (CCI+CBD, SHAM+VEHI, and SHAM+CBD) in the same lineages (Figure 2A–C) and compared to respective baselines. Furthermore, in the lower and higher anxiety lineages (CLF and CHF), the mechanical threshold in CCI+CBD was different compared to SHAM+VEHI and SHAM+CBD on the 23rd day in the same lineage (*p* < 0.0001), as well as when compared with the CCI+CBD group of the CTL lineage (*p* < 0.0001, Figure 2C). Furthermore, CBD treatment completely reversed the mechanical allodynia in the CTL lineage since the mechanical threshold of the CCI+CBD groups on the 23rd day did not differ from the respective baseline (Figure 2A,B). In contrast, this reversal was partial in the CLF and CHF lineages because the mechanical threshold of the CCI+CBD group on the 23rd day differed from the respective baseline and the mechanical threshold on the 13th day (*p* < 0.0001, Bonferroni test, Figure 2C).

### 2.2. Thermal Sensitivity to Cold

The data analysis from the acetone evaporation test showed that CBD treatment partially reduced the cold allodynia independent of the lineages (Figure 2D–F). Repeated-measures analysis of variance (ANOVA) applied in all the experimental groups revealed a significant difference in the lineage, condition, treatment, and time (Table 1). The Bonferroni post hoc test evidenced a significant difference (*p* < 0.0001) in the cold score on the 13th day in the CCI groups compared to the respective SHAM groups (Figure 2D–F) and respective baselines. CBD chronic treatment reduced this cold allodynia in all lineages since the cold score on the 23rd day in the CCI+VEHI groups was different (*p* < 0.0001, Bonferroni test) when compared to other groups (CCI+CBD, SHAM+VEHI, and SHAM+CBD) in the same lineage (Figure 2D–F) and compared to respective baselines. However, this effect was partial because, on the 23rd day, the cold score of the CCI+CBD groups of all lineages differed from SHAM+VEHI and SHAM+CBD (*p* < 0.0001, Bonferroni test, Figure 2D–F) and their respective baselines (*p* < 0.0001, Bonferroni test). The CLF and CHF lineages showed a significant difference when compared with the CTL lineage on the 23rd after surgery in CCI conditions treated with CBD, and CHF+CCI treated with VEHI was different from CLF and CTL in the same condition and treatment (*p* < 0.0001, Bonferroni test, Figure 2D–F).

### 2.3. Thermal Sensitivity to Heat

The present data showed that CCI reduced the hot plate latency at the 13th day independent of lineage (CCI groups were different from SHAM groups in all lineages), and the CBD treatment recovery of the thermal sensitivity evaluated on the 23rd experimental day (Figure 2G–I). Statistical analysis (repeated-measures analysis of variance, ANOVA) evidenced a significant difference in the lineage, condition, treatment, and time (Table 1). The results revealed no difference among the lineages and the hot plate latency in the CCI groups at baseline and on the 13th day. Further, on the 13th day, the hot plate latency of the CCI groups (CCI+CBD and CCI+VEHI) differed from the respective SHAM groups and baseline of all lineages. Additionally, in the CCI+VEHI groups (in all lineages), the hot plate test baseline latencies were different when compared to latencies at the 13th and 23rd days (*p* < 0.05, Bonferroni test, Figure 2G–I, all lineages). On the 23rd day in the CCI groups that received CBD treatment (CCI+CBD), the latencies were different compared to the respective latencies on the 13th day and did not differ from respective baselines (*p* < 0.05, Bonferroni test, Figure 2G–I) in all lineages. However, on the 23rd day, in the CHF lineage, the SHAM+VEHI and SHAM+CBD groups were different (*p* < 0.001, Bonferroni test) compared the same condition and treatment to CLF and CTL lineages (Figure 2G–I) and compared to hot plate latency in the 13th day and the baseline (*p* < 0.05, Figure 2I).

### 2.4. The Open Field Test

The evaluation of OFT measurements with the two-way ANOVA was carried out to analyze the % time in the center and the total crossings (Figure 3). These results can be observed in Table 2. Analyses were performed for all groups, and OFT was performed on day 22 after surgery (SHAM and CCI). A two-way ANOVA applied in the % time in the center revealed a significant difference in the lineage, condition, and treatment (Table 2). The results showed that CCI promoted a reduction of the % time in the center in the CCI+VEHI groups compared to respective SHAM+VEHI groups (*p* < 0.05, Bonferroni test, Figure 3A). The CBD treatment reversed the reduction of % time in center induced by CCI in the CTL and CHF lineages since CCI+CBD groups were different compared to CCI+VEHI groups in these lineages (CTL and CHF, *p* < 0.05, Bonferroni test, Figure 3A). The % time in center in the groups SHAM+VEHI and CCI+VEHI of the CLF lineage were different compared to the same condition and treatment of the CHF lineage (SHAM+VEHI and CCI+VEHI groups, Figure 3B). Moreover, in the CLF lineage, the CCI+VEHI group was not different from CCI+CBD (Figure 3A).

A two-way ANOVA applied in the total of crosses revealed a significant difference in the lineage, condition, and interaction (Table 2). In the SHAM condition, there was no difference between the lineages in the VEHI treatment (*p* < 0.05, Bonferroni test, Figure 3B). Further, in the SHAM condition, CBD treatment promoted an increase in the number of crosses in the CLF lineage compared to SHAM+VEHI at the same lineage (*p* = 0.011, Bonferroni test, Figure 3B) and compared to CTL and CHF lineages at the same condition and treatment (*p* < 0.05, Bonferroni test). Considering the CCI+VEHI groups, the CLF lineage was different compared to CTL and CHF lineages (*p* < 0.05, Bonferroni test).

## 3. Discussion

The present results showed that neuropathic pain induced by CCI promoted a mechanical and thermal threshold reduction in all lineages. These alterations were accompanied by a reduction in the % time in the center, suggesting an anxiogenic-like effect associated with persistent pain. The chronic treatment with CBD produced an anti-allodynic effect in mechanical and cold sensitivity and an anti-hyperalgesic effect on the heat stimulus (Figure 2A–I). In this way, the CBD treatment reversed the anxiogenic-like effect in the CTL and CHF lineages (Figure 3A).

Previous experimental studies testing CBD in chronic pain conditions have shown the anti-allodynic effect, the reduction of hyperalgesia [11,41], and its anxiolytic effect [18]. Our data revealed that treatment of CBD produced different effects on neuropathic pain in individuals with distinct anxiety levels (CLF, CTL, CHF lineages). The lineages did not show basal differences, confirming that nociceptive stimuli did not select these lineages [36,42]. Moreover, a lower efficacy of CBD was observed in the CHF+CCI group compared to the CTL+CCI in mechanical and cold thermal sensitivity, which suggests the hypothesis of an elevated level of sensitivity in this lineage. In addition, partial reversal was also evidenced in the CLF rats, maybe due to this lineage having a different anxiety level than CTL and CHF. In addition, greater sensitivity of CHF animals in evaluating the thermal stimulus to heat when in CCI+VEHI on the last day of the protocol reinforces the hypothesis of greater sensitivity in this lineage. The present results corroborate a previous study with the high anxiety behavior (HAB) and low anxiety behavior (LAB) lineages, where an innate fear of the elevated plus-maze phenotype showed increased pain sensitivity in rats of the HAB lineage [43]. As such, the level of anxiety can modify pain sensitivity in animal models.

Previous report developments with Wistar Hannover rats [41] showed that acute CBD treatment at different doses reversed chronic pain and anxiety induced by CCI. Additionally, this study [41] also showed through immunofluorescence the high activation of CB1 and TRPV1 receptors in the areas of the anterior cingulate cortex (ACC), agranular insular cortex (AIC), basolateral amygdala (BLA), ventral (VH), and dorsal (DH) hippocampus with CBD treatment (at the dose 3 mg/kg). In Carioca rats, the CHF lineage has shown higher Fos protein expression in the BLA and the locus coeruleus (LC) induced by contextual fear compared to the other lineages [42]. Taken together, these morphofunctional differences observed in structures involved in the analgesic and anxiolytic effects of CBD, at least in the BLA and the hippocampus region, may partly explain the different effects of CBD in our protocols.

It is important to note that the SHAM group showed an elevated thermal threshold in the hot plate apparatus of the CHF lineage that could be associated with the facilitation of aversive learning, where, from the start, these animals are selected based on the defensive behavior of freezing using a noxious stimulus [35,44,45]. To extend this hypothesis of learning, a study with the opiate antagonist naltrexone [46] showed increased learning in mice on the hot plate due to the drug’s effect on memory consolidation, regardless of their level of anxiety. These results corroborate the perspective suggested in the present study. Further, this condition of learning facilitation was observed only in the CHF lineage. It is still important to point out that the CHF was selected for the freezing behavior and that the absence of behavior on the hot plate keeps the animal on the apparatus for a longer time, resulting in a higher threshold. It was possible to observe this lineage in the SHAM condition (Figure 2I). The use of CBD did not show a difference in this SHAM group between treatments in CHF animals; the (CHF) SHAM+CBD were different when compared with the baseline, 13th, and 23rd days. In addition, the result was different in VEHI of CHF about the level presented by the CLF and CTL on CBD treatment (Figure 2G–I). Posteriorly, the aversive learning in these animals will be better investigated through CBD treatment in the model corresponding to their selective breeding [34,36,45]. Nonetheless, CBD also had an overall anxiolytic effect in aversive learning tests [47,48,49].

Moreover, on the neural basis of chronic pain, the literature has already established that the LC is a structure that facilitates the transition from acute to chronic pain and the maintenance of chronic pain, containing projections for structures such as the hippocampus, amygdala, and hypothalamus [50]. Furthermore, a previous study that used Fos immunochemistry to investigate changes in neural activity in different brain structures among CHF and CLF rats when they were exposed to contextual cues that were previously associated with footshock has shown a high activation of c-Fos in the LC of CHF animals compared to CLF [42]. Following this perspective, there is a possibility that the LC is a structure in this animal model of anxiety that contains essential processes in pain modulation and aversive learning observed. For a better discussion, further studies are needed to clarify this hypothesis.

Beyond that, there was no difference in locomotion among lineages in the SHAM groups that did not receive CBD in OFT (Figure 3B). The OFT test evaluates the locomotion [51] and the level of anxiety [52] (innate) of these animals. In animals CCI+VEHI, the chronic pain in % time in the center showed an increased level of anxiety in all lineages (Figure 3A). Furthermore, the use of CBD in CTL and CHF lineages (CCI condition) had an anxiolytic effect, still when the CHF rats were in comorbidity with chronic pain (Figure 3A). Studies with humans and non-humans confirm the increase in anxiety in chronic pain [43,53,54,55], which suggests that anxiety in these lineages also intensified, and this was evidenced in the difference between CHF+CCI+VEHI and CLF+CCI+VEHI (Figure 3A). However, CBD showed an anxiolytic effect in a comorbid condition. Therefore, it reduces the type of anxiety analyzed in the OFT associated with an innate fear, which could differ from the anxiety associated with aversive learning that matches the phenotype of this animal model of anxiety.

On the other hand, the CLF lineage in both SHAM (treated with CBD) and CCI conditions showed higher locomotion when compared to the CTL and CHF lineages. However, in the SHAM condition, the CLF treated with CBD presented higher locomotion than other groups of the same lineage (Figure 3B). Additionally, studies showed that lineages selected from a lower level of anxiety are associated with impulsive behaviors [56,57]. The HAB and LAB lineages mentioned above have high locomotion results in the less anxious lineage [58]. This behavior has been previously observed in other studies with the animal model of Carioca [37,40,59]. These results suggest that a more specific investigation is necessary to evaluate a possible attention deficit hyperactivity disorder model in this lineage (CLF), which would imply another condition of comorbidity and also could explain the partial recovery in mechanical and cold sensitivity in this lineage (Figure 3A,D).

Further, some studies noted the CBD effect increases locomotion [58,60,61], while others do not show this difference [58,62]. A study that also evaluated the mobility of Carioca rats in the OFT reaffirms the high mobility of CLF rats [40]. However, the mechanism underlying this elevated locomotion effect of CLF rats remains to be determined [59].

In summary, the results obtained in this study demonstrate the effectiveness of systemic treatment with CBD in parameters of sensory-discriminative aspects with chronic pain independent of the lineage. Chronic treatment with CBD in CLF, CTL, and CHF lineages had anti-allodynic, anti-hyperalgesic, and anxiolytic effects observed, mainly in the condition of comorbid chronic pain in a GAD model. Despite this, the CLF and CHF lineages had a lower recovery in mechanical and thermal sensitivity in CBD treatment compared to the CTL rats. The CHF lineage presented a higher sensitivity and anxiety in chronic pain than other lineages.

## 4. Materials and Methods

### 4.1. Animals

In this study, 96 Wistar rats of F34 and F35 outbreeding generation of the CLF (*n* = 32), CHF (*n* = 32), and control rats CTL (*n* = 32) lineages were used, as described previously [35], from the Animal Facility of the Behavioral Neuroscience Laboratory (LANEC) of the Pontifical Catholic University of Rio de Janeiro (PUC—Rio, Rio de Janeiro, Brazil). At weaning (postnatal day 21), rats of the same group were housed together (six animals per cage) in polypropylene 35 × 19 × 25 cm lined with wood shavings in an environment with controlled temperature (24 °C ± 1 °C) and 12-h light–dark cycle, and water and food ad libitum. The rats were transported by the airline at one month of age to the animal facility of the Department of Psychology at the Faculty of Philosophy, Sciences, and Letters of Ribeirão Preto, and composed of the offspring of randomized crossbreeding populations. To control for litter effects, the rats were randomly selected from a minimum of five different litters per lineage and animals in the control group. After the habituation period (30 days) maintained under the same conditions, the rats were submitted to different treatments. At the beginning of the experiments, the rats were 8 to 10 weeks old, weighing ± 220 g. A trained observer, blind to the experimental group of the animal, performed the protocol procedures. All experimental protocols were carried out under regulations and care in the use of laboratory animals, according to the recommendations of the Conselho Nacional de Controle de Experimentação Animal—Ministério da Ciência e Tecnologia, Brazil, and received the approval of the Committee of Ethics in Animal Use of the University of São Paulo of Ribeirão Preto campus (protocol number 2019.1.833.59.2). G*Power 3.1.7 software and previous studies [37,39,40] were used to determine the sample size. All efforts were made to minimize animal suffering and to use as few animals as possible. The ARRIVE guidelines (Animal Research, reporting of Vivo Experiments) were followed.

### 4.2. Chronic Constriction Injury of the Ischiatic Nerve (CCI) and Control Group (SHAM)

Peripheral neuropathy was induced by ligation of the right ischiatic nerve (sciatic nerve), according to a method previously described by Bennett & Xie [63] and modified by Dias et al. [64]. For this, the rats were anesthetized with the association of 10% ketamine hydrochloride (75 mg/kg, Cetamin, Syntec, Barueri—SP/Brazil) and 2% xylazine hydrochloride (10 mg/kg, Xilazin, Syntec, Barueri—SP/Brazil), administered in the hind limb intramuscularly. After being anesthetized, the animals were placed on a surgical table (dorsal position), and their limbs were immobilized with tape. An internal trichotomy was performed on the hind paw, and the area was disinfected with iodopovidone (Rioquímica, São José do Rio Preto—SP/Brazil). A 1.5 cm incision was made on the inner thigh side, reaching the muscular fascia to separate the gluteal and biceps femoris muscles, exposing the ischiatic nerve. Dissection was performed at the level of the small trochanter of the femur in length, measuring around 8 mm from the proximal ischiatic nerve. The nerve was constricted using a 4-0 chromed Catgut (Bioline—Fios cirúrgicos, Anápolis—GO/Brazil) thread and transfixed with the same wire using a 3/8 cutting mini-needle. The ischiatic nerve was transfixed 3/4 of its diameter and constricted at one point. Then, it was replaced in its original location, the muscles released, and the skin sutured with cotton thread. The false-operated control group (SHAM) only exposed the ischiatic nerve. After the completion of the ligation or simple exposure of the nerve, the epithelial tissue was sutured with 2.0 silk thread. The surgery was performed between 11 a.m. and 4 p.m. A 70% alcohol was used to clean the table, and GermeRio (Rioquímica, São José do Rio Preto—SP/Brazil) was used for disinfecting the surgical instrumentation at each surgery.

### 4.3. Treatment with CBD

The CBD (99.6% purity; BSPG, Pharm, Sandwich, UK) was diluted in vehicle solution (98% saline at 0.015 M and 2% Tween 80) and administered in a volume of 1 mL/kg intraperitoneally from the 14th day after the surgery for constriction of the ischiatic nerve or not (SHAM group). The 5 mg/kg [17,65,66] dose was administered daily for 10-day chronic treatment. A daily administration was performed considering eliminating the time and half-life of CBD to promote the drug’s maintenance in the body [67]. The experimental groups were evaluated in nociceptive tests 1 h after CBD administration on the 23rd day. This time interval was based on CBD’s plasma absorption time [68].

### 4.4. Mechanical Sensitivity

The paw withdrawal reflex evaluated mechanical sensitivity while applying the mechanical stimuli (grams). For this, rats were placed individually in the testing chamber for a 30 min adaptation period. After this time, increasing progressive forces from the filament (0.5 mm in diameter) of an electronic von Frey aesthesiometer (Insight Instruments, Ribeirão Preto, São Paulo, Brazil) were applied to the hind paw plantar surface until the paw was withdrawn [69]. The test sessions were performed before surgery (baseline) and 13 and 23 days after surgery (CCI or SHAM), where the 23rd day was the 10th day of treatment with CBD or VEHI. The paw withdrawal threshold of each animal was calculated as the mean of three values obtained in each session [69,70]. All the von Frey tests were carried out between 9 a.m. and 1 p.m., with 10 min habituation in the room before the test, with room lighting in the 60-lux range, and cleaning of the apparatus with 20% alcohol and paper towel.

### 4.5. Thermal Sensitivity to Cold

The acetone evaporation test evaluated thermal sensitivity to cold in Carioca’s lineages [69]. For this purpose, the rats were kept in individual acrylic boxes with a 5 mm floor corresponding to a mesh net made of 1 mm non-malleable wire. After habituation (15 min), a jet of 100 μL of acetone was instilled into the animal’s right hind paw with an insulin syringe at approximately 5 mm from the paw with peripheral neuropathy through the mesh of the observation box. After 40 s of application of the stimulus, the behavior was evaluated for 2 min using the following score: 0 (no stimulus-response); 1 (quick withdrawal or paw movement); 2 (repeated hind paw movement); and 3 (repeated hind paw movement and licking the paw). The nociceptive response was considered the score to a cold stimulus. The increase or decrease of the values found about the baseline was interpreted as allodynia or cold hyperalgesia, respectively. Three evaluations were performed in each test session with random intervals of 5–10 min between stimuli. Test sessions were performed before surgery (baseline) to induce peripheral neuropathic pain and on days 13 and 23 after CCI or SHAM surgery, where the 23rd day was the 10th day of treatment with CBD or VEHI. The animals were evaluated in the von Frey test and the acetone evaporation test after 10 min interval. All the acetone evaporation tests were conducted between 9 a.m. and 1 p.m., with room lighting in the 60-lux range and cleaning of the apparatus with 20% alcohol and a paper towel.

### 4.6. Hot Plate Test

To assess thermal sensitivity to heat, the time that the animals remained on a heated metallic surface (50 ± 1 °C) until they reacted to the stimulus was measured. At the beginning of the test, the animals were placed on the heated plate, and the response to the thermal stimulus (withdrawal or licking of the hind or forepaws) was recorded. A cut-off time of 30 s was adopted to avoid possible injuries to the animal’s paw due to exposure to harmful thermal stimuli for a long time [69]. The baseline was obtained by the mean of three measurements in the hot plate before the surgery to induce peripheral neuropathy pain. The test sessions were performed on days 13 and 23 after the surgery, where the 23rd day was the 10th day of treatment with CBD or VEHI. The animals were evaluated in the von Frey test, followed by the acetone evaporation test after the 10 min interval and the hot plate test after 10 min interval. The hot plate tests were conducted between 9 a.m. and 1 p.m., with room lighting in the 60-lux range and cleaning of the apparatus with 20% alcohol and a paper towel.

### 4.7. Locomotion Activity

The rats were submitted to the open field test (OFT) [51] on the 22nd experimental day to investigate locomotion pattern and emotional behavior. The OFT is one of the main animal models for studying anxiety-like behavior, capable of detecting changes in the emotional and exploration pattern and being sensitive to classic anxiolytics [52]. The open field arena (60 cm in diameter circle and 50 cm high) can be divided into internal and external quadrants. The time spent on the periphery is directly related to the animal’s aversion and anxiety [52]. In this study, the animals were placed in the apparatus center for 5 min for behavioral analysis. The parameters used to assess the animal’s exploration pattern in the OFT were: time spent in the center and the number of crossings. Finally, the number of crossings was registered to investigate possible changes in evaluating the nociceptive tests since the evaluation indexes depend on motor responses. The OFT was carried out between 11 a.m. and 1 p.m., with room lighting in the 60-lux range and cleaning of the apparatus with 20% alcohol and a paper towel. The tests were recorded by a video camera linked to a monitor outside the experimental room; this allowed the recordings to be analyzed later.

### 4.8. Statistical Analysis

The data are represented as mean ± standard error of the mean (SEM). The comparisons of the nociceptive tests (von Frey, acetone, and hot plate tests) were carried out using repeated-measures analysis of variance (ANOVA) considering lineage (CLF, CTL, or CHF), condition (SHAM or CCI), treatment (VEHI or CBD), and time as factors, followed by the Bonferroni test for comparison among groups to all the tests. Therefore, to analyze the OFT, a one-way ANOVA was used followed by a Bonferroni test for multiple comparisons. The level of significance was set at *p* < 0.05 for all analyses. All statistical analyses were performed using IBM SPSS 23 software (SPSS Inc., Chicago, IL, USA), and the figures were prepared using Prism software (version 9.0, Graph Pad Software).

## 5. Conclusions

The results obtained in this study demonstrated the effectiveness of systemic treatment with CBD on parameters of discriminative sensory aspects with chronic pain, regardless of lineage in this animal model of anxiety. Furthermore, the results of the present study suggest that the mechanical antiallodynic and cold thermal effects of CBD may depend on the level of anxiety.

## Figures and Tables

**Figure 1 pharmaceuticals-16-01003-f001:**
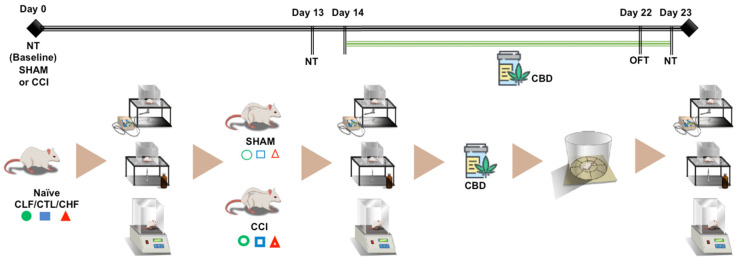
Experimental design, the apparatus of the experimental tests, and the treatment drug in the periods in which they were carried out. Carioca high- (CHF), control (CTL), and low- (CLF) freezing rats were evaluated three times throughout the chronic neuropathic pain protocol, at baseline, on the 13th day, and on the 23rd day after surgery of sciatic nerve constriction (CCI or SHAM). The tests that evaluated the sensorial-discriminative aspects follow, in sequence, the von Frey test, the acetone test, and the hot plate test. The animals were treated with cannabidiol (CBD, 5 mg/kg, daily) or VEHI from the 14th to the 23rd day, and on the 22nd day, the open field test (OFT) was performed. Abbreviations: NT: nociceptive tests.

**Figure 2 pharmaceuticals-16-01003-f002:**
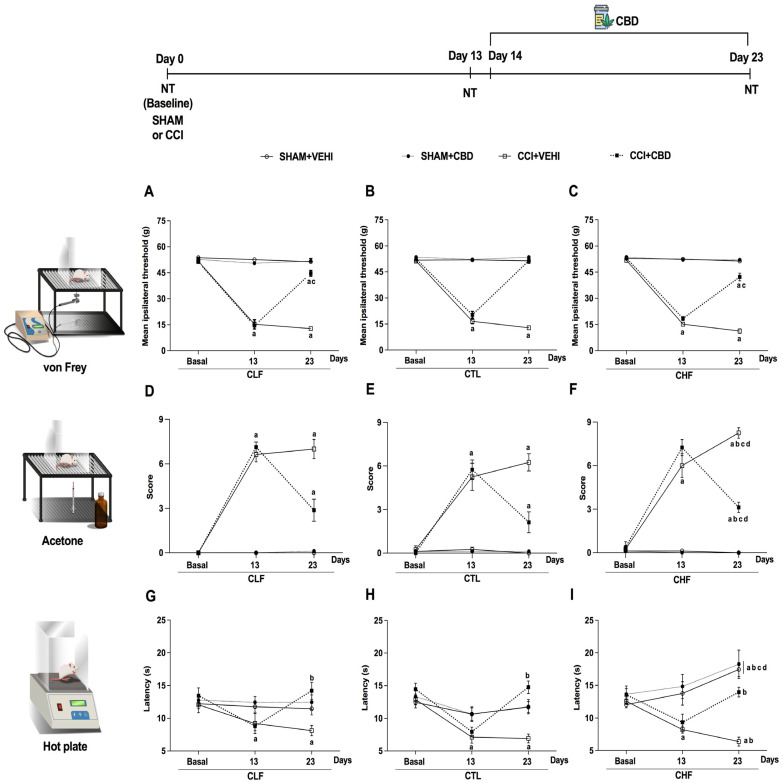
Chronic treatment with cannabidiol (CBD) reverses the mechanical allodynia, reduces thermal allodynia and hyperalgesia induced by chronic constriction injury of the ischiatic nerve (CCI) on Carioca low-conditioned freezing (CLF), high-conditioned freezing (CHF), or control (CTL) rats. Mechanical sensitivity was evaluated by the von Frey test (**A**–**C**); thermal sensitivity was evaluated in the acetone (**D**–**F**) and hot plate tests (**G**–**I**). The mechanical or thermal thresholds were evaluated at the baseline (Basal), after CCI or simulated surgery (SHAM, day 13), and after the treatment with CBD (5 mg/kg/day, i.p., for ten days, starting at the 14th experimental day) or vehicle (VEHI). a *p* < 0.05, Bonferroni test compared with its respective baseline and different treatments at the same lineage; b *p* < 0.05, Bonferroni test compared with day 13 and day 23 at the same treatment; c *p* < 0.05, Bonferroni test compared with CTL lineage at the same treatment; d *p* < 0.05, Bonferroni test compared with the CLF lineage at the same treatment. *N* = 8 animals per group. Abbreviations: NT: nociceptive tests.

**Figure 3 pharmaceuticals-16-01003-f003:**
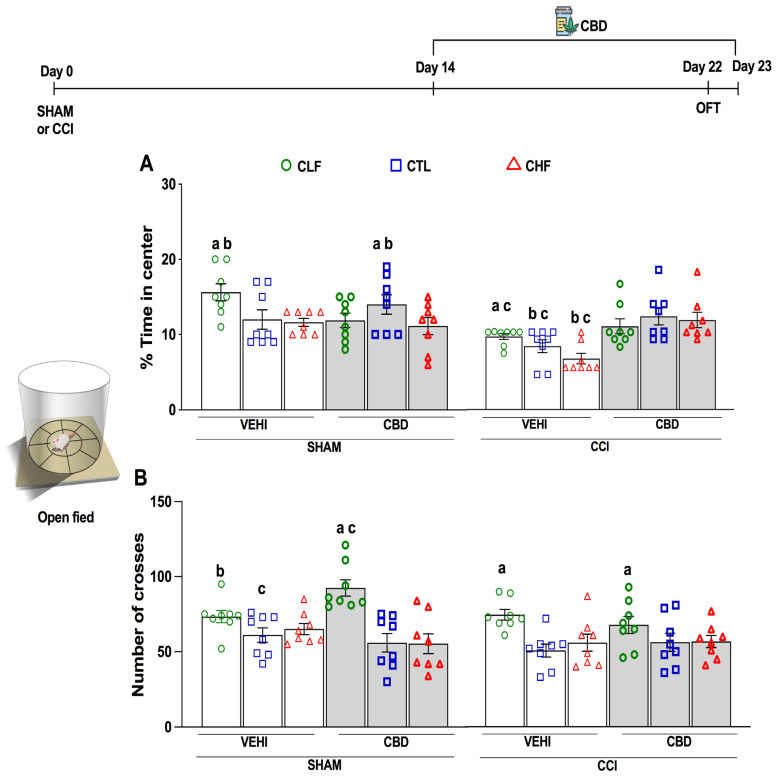
Chronic treatment with cannabidiol (CBD) prevented the anxiogenic effect induced by chronic constriction injury of the ischiatic nerve (CCI) on Carioca low-conditioned freezing (CLF), high-conditioned freezing (CHF), or control (CTL) rats dependent on the lineage evaluated in the open field test (OFT). The measures analyzed were the percentage (%) of time spent in the center (**A**), and the total number of crossing (**B**) behaviors. The OFT was performed for 5 min, 22 days after chronic constriction injury of the ischiatic nerve (CCI) or simulated surgery (SHAM) and after ten days of the treatment with CBD (at the doses 5 mg/kg/day, i.p., starting at the 14th experimental day) or vehicle (VEHI). a *p* < 0.05 Bonferroni test compared the among lineages (CLF, CTL, CHF) in the same condition and treatment; b *p* < 0.05 Bonferroni test compared the same lineage and condition, but different treatment (VEHI or CBD). c *p* < 0.05 Bonferroni test compared the same lineage and treatment, but different condition (SHAM or CCI). Data represent means ± standard errors. *N* = 8 for each experimental group.

**Table 1 pharmaceuticals-16-01003-t001:** Original data of nociceptive tests (von Frey, acetone, and hot plate) in the lineages (CFL, CTL, and CHF) submitted to repeated-measures analysis of variance (ANOVA), followed by the post hoc Bonferroni test. The level of significance was *p* < 0.05 for all analyses.

Nociceptive Tests	Lineage Factor	Condition Factor	Treatment Factor	Time Factor	Interaction
	(CLF vs. CTL vs. CHF)	(SHAM vs. CCI)	(VEHI vs. CBD)	(Basal × 13th × 23th)	(Lineage vs. Condiction vs. Treatment)
von Frey	*F(_**4**, **168**_)* = 6.409	*F(_**2**, **168**_)* = 547.176	*F(_**2**, **168**_)* = 312.546	*F(_**2**, **168**_)* = 657.563	*F(_**4**, **168**_)* = 2.187
	*p* < 0.05	*p* < 0.0001	*p* < 0.0001	*p* < 0.0001	*p* < 0.05
Acetone	*F(_**4**, **168**_)* = 1.811	*F(_**2**, **168**_)* = 247.434	*F(_**2**, **168**_)* = 43.275	*F(_**2**, **168**_)* = 244.175	*F(_**4**, **168**_)* = 0.368
	*p* > 0.0001	*p* < 0.0001	*p* < 0.0001	*p* < 0.0001	*p* > 0.0001
Hot Plate	*F(_**4**, **168**_)* = 5.137	*F(_**2**, **168**_)* = 17.451	*F(_**2**, **168**_)* = 10.532	*F(_**2**, **168**_)* = 29.308	*F(_**4**, **168**_)* = 0.479
	*p* = 0.001	*p* < 0.0001	*p* < 0.0001	*p* < 0.0001	*p* > 0.05

**Table 2 pharmaceuticals-16-01003-t002:** Original data of the open field test (OFT) with a two-way ANOVA, followed by the Bonferroni test. The level of significance was *p* < 0.05 in all analyses.

OFT	Lineage Factor	Condition Factor	Treatment Factor	Interaction
	(CLF vs. CTL vs. CHF)	(SHAM vs. CCI)	(VEHI vs. CBD)	(Lineage vs. Condiction vs. Treatment)
% time in center	*F(_**2.84**_)* = 3.127	*F(_**1.84**_)* = 8.156	*F(_**1.84**_)* = 6.038	*F(_**2.84**_)* = 1.156
	*p* < 0.05	*p* < 0.05	*p* < 0.05	*p* > 0.05
Total of crosses	*F(_**2.84**_)* = 20.519	*F(_**1.84**_)* = 5.445	*F(_**1.84**_)* = 0.045	*F(_**2.84**_*) = 4.248
	*p* < 0.0001	*p* < 0.0001	*p* > 0.05	*p* < 0.05

## Data Availability

Data is available within article.

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
