# Peer review of "Systemic Chronic Treatment with Cannabidiol in Carioca High- and Low-Conditioned Freezing Rats in the Neuropathic Pain Model: Evaluation of Pain Sensitivity"

_pharmaceuticals, 2023, doi:10.3390/ph16071003_

Round 1

Reviewer 1 Report

This research is to investigate the weather chronic systemic treatment with CBD modulates pain sensitivity in rats from an animal anxiety model submitted to a neuropathic pain model. The development of approaches aimed at treating neuropathic pain is undoubtedly an urgent problem in modern neurobiology and medicine. The presented study is interesting, high-quality and well-researched. However, I have a few non-principal questions:

 1. It is better to use rats at the age of 12 weeks for the formation of a model of neuropathic pain, rats at the age of 8-10 weeks are too young.

2. I would like more information in the manuscript about the choice of such an experiment design, why was the drug administered only on the 14th day after the operation? The study of animals for the manifestation of tactile allodynia is indeed more appropriate in the later stages after surgery. However, thermal allodynia is more pronounced by the 3rd day after CCI, but these indicators have not been tested.

3. Do the authors think that testing the sensitivity of the injured limb with three tests on the same day may be redundant to obtain correct data?

4. It would be very interesting to see data from animal anxiety testing in the elevated plus maze. Testing in an open field reflects the motor activity of experimental animals. Write the size open field.

In general, the manuscript is very well written and illustrated, I recommend it for publication.

Reviewer 2 Report

First of all, I have found this project interesting, well planed and documented. The text formulation is understandable. I would suggest some minor refinements:

1.       In the introduction please describe better the Carioca lines. Maybe they are well known in Brazil, but if the reader meets those first in this text, they are a little bit obscure. Please describe that they originate from Wistar strain, and what are the basic properties of the two lines. It would be also interesting how large is the breeding colony, because if it is not large enough they could get some inbred properties and drift from the original stain away also in other properties than anxiety. 

2.       For Materials and Methods chapters:

a.       The animal number in line 302 is somehow wrong it should be either 64 without controls or 96 with.

b.       I would suggest to change the adapted word in line 327 to modified since this operation method is not the classical Bennett and Xie CCI type.

3.       For better understanding of the results I would request a strong modification of the significance marking. I think that the main statistical method was perfect, but I would show on the figures the difference in each time point between different group: I mean to compare the effect to sham and CCI only points and not to previous time point date of the group. It would show mostly the same but it would better understandable. One exception could be the hot plate assay results, where the sham lines (especially in the case of CHF) are not horizontal, so it is a more tricky situation. But e.g. in VF test CCI-CBD day 13 is almost the same as CCI-VEH day 23 and CCI-CBD Basal is the same as SHAM on day 23.
On figure 2F for CHF-CCI-CBD day 23 point it is shown that it is significant different to CLF and CTL-CCI-CBD points but it is for me almost unbelievable. Please modify slightly the post hoc test setting and show the significance on day 23 between the 12 groups. This is the most important. All other post hoc results are much less interesting. At least in prism it would be possible.

4.       On Fig. 3 bar graph it would be easier to show the differences by connecting the bar with lines to show which are different compared to others.

5.       The discussion should be revised, because some statements are not fully true or not fully established by the findings. I would be more careful with statements, because the findings are interesting but the differences are not extreme high.

a.       I disagree with the following sentence: “Moreover, the hypothesis of high sensitivity in CHF animals was directly 215 evidenced since an elevated level of sensitivity was observed in all nociceptive tests in 216 CCI+VEHI groups of this lineage (Fig. 2). “ This was not shown. The day 13 points in VF are total practically identical of CCI groups, in acetone test the CTL is slightly lower but CLF and CHF are the same, on HP test it would be surprising to get difference. To support the previous statement the CCI-VEH group results of the 3 lines on day 13 should be compared.

b.       The next sentence “Further, a lower efficacy of CBD was observed 217 in the CHF+CCI group compared to the CTL+CCI.” is also not fully true. It is true for VF, maybe for HP (but the significance symbol is missing…) but not true for acetone.

c.       It should somehow be explained why also in the opposite anxiety line (CLF) had CBD less effect than in CHF.

d.       I also not fully agree with the following sentence (line 296): “Despite this, the CLF and CHF 296 lineages had a lower recovery in mechanical and thermal sensitivity in CBD treatment 297 compared to CTL rats.” It is true for mechanical but it was not shown for thermal tests.

e.       I would suggest also better explain the previous in vitro findings of Carioca rats, because the description is superficial for understanding. Please write down what CO and HO abbreviations are.
